# Design, Synthesis, Mode of Action and Herbicidal Evaluation of Quinazolin-4(3*H*)-one Derivatives Based on Aryloxyphenoxypropionate Motif

Chaochao Wang [1], Ke Chen [2], Na Li [1], Shuyue Fu [1], Pan Li [1], Lusha Ji [1], Guoyun Liu [1], Xuekun Wang [1] and Kang Lei [1,*]

[1]  School of Pharmaceutical Sciences, Liaocheng University, Liaocheng 252000, China;
    kelsey9709@163.com (C.W.); 17863527116@163.com (N.L.); fsyue1213@163.com (S.F.); lipan@lcu.edu.cn (P.L.);
    jilusha@lcu.edu.cn (L.J.); liuguoyun@lcu.edu.cn (G.L.); wangxuekun@lcu.edu.cn (X.W.)
[2]  Medical School, Liaocheng University, Liaocheng 252000, China; chenke7758@163.com
*   Correspondence: leikang@lcu.edu.cn

**Abstract:** To discover new acetyl-CoA carboxylase (ACCase) inhibiting-based herbicides, twenty-nine novel quinazolin-4(3*H*)-one derivatives were designed and synthesized based on the aryloxyphenoxypropionate motif. The bioassay results showed that most of the target compounds showed better pre-emergent herbicidal activity against monocotyledonous weeds in a greenhouse. Especially, when applied at 375 g ha$^{-1}$ under pre-emergence conditions, compound **QPP-7** displayed excellent herbicidal activity against monocotyledonous weeds (i.e., *E. crusgalli*, *D. sanguinalis*, *P. alopecuroides*, *S. viridis*, *E. indica*, *A. fatua*, *E. dahuricu*, *S. alterniflora*) with inhibition rate >90%, and displayed excellent crop safety to *O. sativa*, *T. aestivum*, *G. spp*, and *A. hypogaea.* The study of structure-activity relationship (SAR) revealed that the herbicidal activity of target compounds is strongly influenced by the spatial position of R group and the bulk of $R_1$ group on quinazolin-4(3*H*)-one, and the (R = 6-F, $R_1$ = Me) pattern is confirmed as the optimal orientation. Furthermore, the molecular docking study and the good inhibitory activity of **QPP-7** against *E. crusgalli* ACCase enzyme ($IC_{50}$ = 54.65 nM) indicated that it may be a ACCase inhibitor. Taken together, the present work demonstrated that compound **QPP-7** could serve as a potential lead structure for further developing novel ACCase inhibiting-based herbicide.

**Keywords:** quinazolin-4(3*H*)-one derivatives; synthesis; herbicidal activity; molecular docking; ACCase inhibitor

## 1. Introduction

Herbicides play an important role in weeds control, protecting crops, and increase yields in agriculture. Among the known herbicides, aryloxyphenoxypropionate (APP) are a class of herbicides that inhibit the synthesis of fatty acids and destroy the membrane structure by inhibiting the activity of acetyl-CoA carboxylase in gramineous plants to achieve herbicidal effects [1–4]. Since the first launch of diclofop-methyl in 1971, many of APP herbicides, such as haloxyfop-P-methyl, fluazifop-P-butyl, fenoxaprop-P-ethyl, quizalofop-P-ethyl, have been reached the marketplace (Figure 1A). However, an inevitable problem associated with long-term irrational use of APP herbicides is the reduced efficacy due to weed resistance [5–10]. To overcome this problem, developing APP herbicides with novel structure or improved herbicidal activity is necessary.

Biologically active natural products (NPs) are often served as the lead structures for novel agrochemical discovery in that their advantages associated with unique mode of action, easy degradation, and good environmental compatibility [11–14]. In addition, many of previous works have shown that the introduction of natural active groups is also an efficient method for agrichemical discovery, and a variety of NP-derived pesticides have

been successfully developed and brought to market [15,16]. Quinazolin-4(*3H*)-ones, an important class of *N*-containing heterocyclic compounds based on a benzopyrimidone alkaloid structure, are widely distributed in plants and microorganisms (Figure 1B) [17]. Over the last few decades, natural quinazolin-4(*3H*)-ones have been found to possess a wide range of biological activities such as antifungal, [18] anticancer, [19] antiviral, [20,21] radical-scavenging, [22] antimicrobial, [23] cytotoxicity, [24] and anti-inflammatory, [25] and anti-malaria activities [26]. As such, the quinazolin-4(*3H*)-one skeleton have received considerable attention in recent years and is considered to be a privileged structure for developing drugs and pesticides [27–33]. So far, many of the quinazolin-4(*3H*)-one derivatives have been introduced into the market as drugs or pesticides, such as diproqualone (an anti-rheumatic drug), methaqualone (anti-convulsant drugs), raltitrexed (an anticancer drug), and fluquinconazole (an agricultural fungicide) have been developed to reach the market. Although there are many therapeutic drugs and pesticides based on the quinazolin-4(*3H*)-one skeleton, commercial herbicides based on the quinazolin-4(*3H*)-one skeleton are rarely reported.

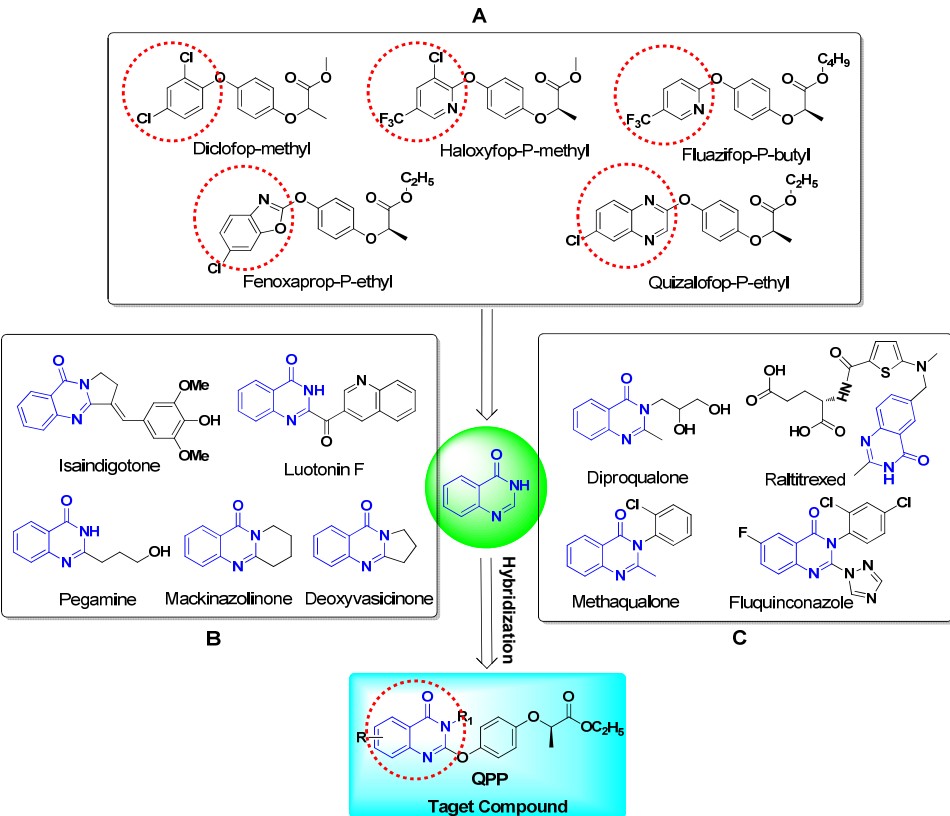

**Figure 1.** Design of target compound **QPP** by molecular hybridization strategy. (**A**)commercial APP herbicides; (**B**) the natural quinazolin-4(*3H*)-ones; (**C**) commercial drugs containing quinazolin-4(*3H*)-one motif.

Based on the above facts, in order to develop novel APP herbicides containing quinazolin-4(*3H*)-one skeleton with commercial potential, we intend to replace the aromatic ring part of the APP herbicides with quinazolin-4-one motif to construct quinazolinone-APP hybrids (Figure 1), which is expected to possess good herbicidal activity. Therefore, as part of our continuous efforts to develop novel structures with potential use as herbicides, [34–37] twenty-nine novel quinazolin-4-one derivatives based on the APP motif were designed, synthesized and tested for herbicidal activity and molecular mode action. To the best of our knowledge, this is the first report on the herbicidal activity of quinazolin-4(*3H*)-one derivatives with an APP motif.

## 2. Materials and Methods

### 2.1. General Information

In most cases, the reagents and solvents, purchased form Energy Chemical or Tokyo Chemical Industry, were analytical grade and used without further purification. Column chromatography purification was carried out using silica gel column chromatography (silica gel 200–300 mesh) (Qingdao Makall Group Co., Ltd., Qingdao, China). $^1$H and $^{13}$C NMR spectrum are obtained at 500 MHz and 125 MHz, respectively, using an AV-500 spectrometer (Bruker, Billerica, MA, USA) in CDCl$_3$ or DMSO-$d_6$ solution with tetramethylsilane (TMS) as the internal standard. The chemical shifts are reported as $\delta$ values relative to TMS. High-resolution mass spectra is conducted using an Ionspec 7.0 T spectrometer (Varian, Palo Alto, CA, USA) by the electrospray ionisation fourier transform ion cyclotron resonance (ESI-FTICR) technique. The crystal structure was determined on a Saturn 724 CCD area-detector diffractometer (Rigaku, Tokyo, Japan). High-performance liquid chromatography (HPLC) data is obtained on a SHIMADZU LC-20AT (Japan).

### 2.2. Chemical Synthesis Procedures

The synthetic pathway used to prepare the target compounds **QPP-1** to **QPP-29** is outlined in Scheme 1. The yields were not optimized.

**Route A**

**1a** R= H; **1b** R= 6-Me; **1c** R= 5-Me; **1d** R= 4-Me; **1e** R= 3-Me; **1f** R= 6-F; **1g** R= 5-F; **1h** R= 4-F; **1i** R= 3-F; **1j** R= 6-Cl; **1k** R= 5-Cl; **1l** R= 4-Cl; **1m** R= 3-Cl; **1n** R= 4,5-2Me; **1o** R= 4,5-2F; **2a/3a** R= H; **2b/3b** R= 5-Me; **2c/3c** R= 6-Me; **2d/3d** R= 7-Me; **2e/3e** R= 8-Me; **2f/3f** R= 5-F; **2g/3g** R= 6-F; **2h/3h** R= 7-F; **2i/3i** R= 8-F; **2j/3j** R= 5-Cl; **2k/3k** R= 6-Cl; **2l/3l** R= 7-Cl; **2m/3m** R= 8-Cl; **2n/3n** R= 6,7-2Me; **2o/3o** R= 6,7-2F; **QPP-1** R= H; **QPP-2** R= 5-Me; **QPP-3** R= 6-Me; **QPP-4** R= 7-Me; **QPP-5** R= 8-Me; **QPP-6** R= 5-F; **QPP-7** R= 6-F; **QPP-8** R= 7-F; **QPP-9** R= 8-F; **QPP-10** R= 5-Cl; **QPP-11** R= 6-Cl; **QPP-12** R= 7-Cl; **QPP-13** R= 8-Cl; **QPP-14** R= 6,7-2Me; **QPP-15** R= 6,7-2F.

**Route B**

**4a/5a/6a** R$_1$= Et; **4b/5b/6b** R$_1$= *n*-Pr; **4c/5c/6c** R$_1$= *n*-Butyl; **4d/5d/6d** R$_1$= *i*-Pr; **4e/5e/6e** R$_1$= Phenyl; **4f/5f/6f** R$_1$= 2-Methylphenyl; **4g/5g/6g** R$_1$= 3-Methylphenyl; **4h/5h/6h** R$_1$= 4-Methylphenyl; **4i/5i/6i** R$_1$= 2-Chlorophenyl; **4j/5j/6j** R$_1$= 3-Chlorophenyl; **4k/5k/6k** R$_1$= 4-Chlorophenyl; **4l/5l/6l** R$_1$= 2-Trifluoromethylphenyl; **4m/5m/6m** R$_1$= 3-Trifluoromethylphenyl; **4n/5n/6n** R$_1$= 4-Trifluoromethylphenyl; **QPP-16** R$_1$= Et; **QPP-17** R$_1$= *n*-Pr; **QPP-18** R$_1$= *n*-Butyl; **QPP-19** R$_1$= *i*-Pr; **QPP-20** R$_1$= Phenyl; **QPP-21** R$_1$= 2-Methylphenyl; **QPP-22** R$_1$= 3-Methylphenyl; **QPP-23** R$_1$= 4-Methylphenyl; **QPP-24** R$_1$= 2-Chlorophenyl; **QPP-25** R$_1$= 3-Chlorophenyl; **QPP-26** R$_1$= 4-Chlorophenyl; **QPP-27** R$_1$= 2-Trifluoromethylphenyl; **QPP-28** R$_1$= 3-Trifluoromethylphenyl; **QPP-29** R$_1$= 4-Trifluoromethylphenyl.

$^a$ Regeant and conditions: (a) Et$_3$N, EtOH, 80 °C, 3 h; (b) SO$_2$Cl$_2$, CHCl$_3$, 60 °C, 2h; (c) K$_2$CO$_3$, CH$_3$CN, 90 °C, 2h.

**Scheme 1.** Synthetic route of preparing target compounds **QPP-1** to **QPP-29**.

#### 2.2.1. General Procedure for the Synthesis of Intermediates **2a–2o** and **5a–5n**

Intermediates **2a–2o** and **5a–5n** were prepared following a reported method [38]. To a 100 mL round-bottom flask was added anthranilic acid **1a** (2.74 g, 20.0 mmol), methyl isothiocyanate (1.61 g, 22.0 mmol), Et$_3$N (2.22 g, 22.0 mmol) and EtOH (30 mL). The reaction mixture was stirred at 80 °C for 3 h. After the reaction cooled to room temperature, the resulting precipitates was filtered, and the solid was washed with 20 mL EtOH/20 mL hexane, and dried to acquire the pure product **2a** as a white solid (3.56 g, yield: 92.7%).

Intermediates **2b–2o** and **5a–5n** were prepared by the similar procedure to **2a**. For data on **2a–2o** and **5a–5n**, see the supporting information.

### 2.2.2. General Procedure for the Synthesis of Intermediates **3a–3o** and **6a–6n**

Intermediates **3a–3o** and **6a–6n** were prepared following a reported method [38]. To a suspension of compound **2a** (1.92 g, 10.0 mmol) in $CHCl_3$ (25 mL) was added $SO_2Cl_2$ (1.46 g, 11.0 mmol). The reaction mixture was stirred at 60 °C for 2 h. After the completion of the reaction, the mixture was cooled to room temperature and diluted with $CH_2Cl_2$ (30 mL). The organic mixture was washed with brine, dried with $Na_2SO_4$, filtered and concentrated to be purified through chromatograph on silica gel using petroleum ether/ethyl acetate (V:V = 20:1) as eluent to give white solid **3a** (1.17 g, yield: 60.3%).

Intermediates **3b–3o** and **6a–6n** were prepared by the similar procedure to **3a**. For data on **3a–3o** and **6a–6n**, see the supporting information.

### 2.2.3. General Procedure for the Synthesis of Target Compounds **QPP-1** to **QPP-29**

Compound **3a** (194 mg, 1.0 mmol) was dissolved in 20 mL acetonitrile followed by addition of (R)-ethyl 2-(4-hydroxyphenoxy)propanoate (210 mg, 1.0 mmol) and potassium carbonate (207 mg, 1.5 mmol). The reaction mixture was stirred at 90 °C for 2 h. After the reaction was completed according to TLC detection, the solvent was removed under reduced pressure. The residue was purified through chromatograph on silica gel using petroleum ether/ethyl acetate (V:V = 10:1) as eluent to give target compound **QPP-1** as a white solid (317 mg, yield: 86.2%). Target compounds **QPP-2** to **QPP-29** were prepared by the similar procedure to **QPP-1**.

(R)-ethyl 2-(4-((3-methyl-4-oxo-3,4-dihydroquinazolin-2-yl)oxy)phenoxy)propanoate (**QPP-1**): white solid, yield 86.2%, m.p. 78–80 °C; [1]H NMR (500 MHz, $CDCl_3$) δ: 8.21 (dd, J = 8.0, 1.3 Hz, 1H), 7.62–7.58 (m, 1H), 7.38–7.30 (m, 2H), 7.20–7.12 (m, 2H), 6.98–6.88 (m, 2H), 4.76 (q, J = 6.8 Hz, 1H), 4.31–4.20 (m, 2H), 3.70 (s, 3H), 1.65 (d, J = 6.8 Hz, 3H), 1.28 (t, J = 7.1 Hz, 3H); [13]C NMR (125 MHz, $CDCl_3$) δ: 172.1, 163.1, 155.4, 152.6, 146.6, 145.9, 134.3, 127.1, 126.0, 124.9, 122.7, 118.9, 115.9, 73.2, 61.4, 28.8, 18.6, 14.2; HRMS, m/z calcd. for $C_{20}H_{21}N_2O_5^+$ [M + H]$^+$ 369.1445, found 369.1452.

(R)-ethyl 2-(4-((3,5-dimethyl-4-oxo-3,4-dihydroquinazolin-2-yl)oxy)phenoxy)propanoate (**QPP-2**): white solid, yield 85.1%, m.p. 101–104 °C; [1]H NMR (500 MHz, $CDCl_3$) δ: 7.45–7.38 (m, 1H), 7.19–7.13 (m, 3H), 7.06 (d, J = 7.3 Hz, 1H), 6.96–6.91 (m, 2H), 4.76 (q, J = 6.8 Hz, 1H), 4.30–4.18 (m, 2H), 3.63 (s, 3H), 2.85 (s, 3H), 1.65 (d, J = 6.8 Hz, 3H), 1.28 (t, J = 7.1 Hz, 3H); [13]C NMR (126 MHz, $CDCl_3$) δ: 172.1, 163.5, 155.3, 152.2, 148.0, 145.9, 141.2, 133.3, 127.7, 124.2, 122.7, 117.4, 115.9, 73.2, 61.4, 28.6, 22.9, 18.6, 14.2; HRMS, m/z calcd. for $C_{21}H_{23}N_2O_5^+$ [M + H]$^+$ 383.1601, found 383.1612.

(R)-ethyl 2-(4-((3,6-dimethyl-4-oxo-3,4-dihydroquinazolin-2-yl)oxy)phenoxy)propanoate (**QPP-3**): white solid, yield 83.3%, m.p. 91–94 °C; [1]H NMR (500 MHz, $CDCl_3$) δ: 7.99 (d, J = 0.8 Hz, 1H), 7.41 (dd, J = 8.3, 2.0 Hz, 1H), 7.26 (d, J = 8.6 Hz, 1H), 7.19–7.13 (m, 2H), 6.99–6.90 (m, 2H), 4.75 (q, J = 6.8 Hz, 1H), 4.27–4.23 (m, 2H), 3.68 (s, 3H), 2.42 (s, 3H), 1.64 (d, J = 6.8 Hz, 3H), 1.28 (t, J = 7.1 Hz, 3H); [13]C NMR (125 MHz, $CDCl_3$) δ: 172.1, 163.1, 155.4, 152.1, 145.9, 144.4, 135.8, 134.9, 126.5, 125.8, 122.7, 118.6, 116.7, 116.1, 115.9, 73.2, 61.4, 28.8, 21.1, 18.6, 14.2; HRMS, m/z calcd. for $C_{21}H_{23}N_2O_5^+$ [M + H]$^+$ 383.1601, found 383.1607.

(R)-ethyl 2-(4-((3,7-dimethyl-4-oxo-3,4-dihydroquinazolin-2-yl)oxy)phenoxy)propanoate (**QPP-4**): white solid, yield 81.6%, m.p. 87–90 °C; [1]H NMR (500 MHz, $CDCl_3$) δ: 8.09 (d, J = 8.1 Hz, 1H), 7.20–7.09 (m, 4H), 6.98–6.87 (m, 2H), 4.76 (q, J = 6.8 Hz, 1H), 4.25 (qd, J = 7.1, 1.6 Hz, 2H), 3.68 (s, 3H), 2.39 (s, 3H), 1.65 (d, J = 6.8 Hz, 3H), 1.29 (t, J = 7.1 Hz, 3H); [13]C NMR (125 MHz, $CDCl_3$) δ: 172.1, 162.9, 155.4, 152.7, 146.7, 145.9, 145.3, 126.9, 126.5, 125.8, 122.7, 116.5, 115.9, 73.2, 61.4, 28.7, 21.8, 18.6, 14.2; HRMS, m/z calcd. for $C_{21}H_{23}N_2O_5^+$ [M + H]$^+$ 383.1601, found 383.1612.

(R)-ethyl 2-(4-((3,8-dimethyl-4-oxo-3,4-dihydroquinazolin-2-yl)oxy)phenoxy)propanoate (**QPP-5**): white solid, yield 80.3%, m.p. 70–72 °C; [1]H NMR (500 MHz, $CDCl_3$) δ: 8.05 (d, J = 7.9 Hz, 1H), 7.45 (d, J = 7.2 Hz, 1H), 7.24–7.16 (m, 3H), 7.01–6.90 (m, 2H), 4.77 (q, J = 6.8 Hz, 1H), 4.30–4.14 (m, 2H), 3.69 (s, 3H), 2.24 (s, 3H), 1.65 (d, J = 6.8 Hz, 3H), 1.26 (t, J = 7.1 Hz, 3H); [13]C NMR (125 MHz, $CDCl_3$) δ: 172.1, 163.4, 155.2, 151.6, 146.2, 145.1, 134.8,

134.3, 124.7, 124.5, 122.7, 118.7, 115.7, 73.3, 61.4, 28.7, 18.6, 16.7, 14.1; HRMS, m/z calcd. for $C_{21}H_{23}N_2O_5^+$ [M + H]$^+$ 383.1601, found 383.1610.

(R)-ethyl 2-(4-((5-fluoro-3-methyl-4-oxo-3,4-dihydroquinazolin-2-yl)oxy)phenoxy)propanoate (**QPP-6**): white solid, yield 78.0%, m.p. 82–85 °C; $^1$H NMR (500 MHz, CDCl$_3$) δ: 7.52–7.48 (m, 1H), 7.17–7.11 (m, 3H), 6.99–6.91 (m, 3H), 4.76 (q, *J* = 6.8 Hz, 1H), 4.32–4.18 (m, 2H), 3.66 (s, 3H), 1.65 (d, *J* = 6.8 Hz, 3H), 1.28 (t, *J* = 7.1 Hz, 3H); $^{13}$C NMR (125 MHz, CDCl$_3$) δ: 172.0, 161.52 (d, *J* = 265.1 Hz), 159.9 (d, *J* = 4.0 Hz), 155.5, 153.2, 148.6, 145.7, 134.5 (d, *J* = 10.5 Hz), 122.7, 121.9 (d, *J* = 4.0 Hz), 115.9, 111.6 (d, *J* = 20.7 Hz), 73.2, 61.4, 28.5, 18.6, 14.2; HRMS, m/z calcd. for $C_{20}H_{20}FN_2O_5^+$ [M + H]$^+$ 387.1351, found 387.1355.

(R)-ethyl 2-(4-((6-fluoro-3-methyl-4-oxo-3,4-dihydroquinazolin-2-yl)oxy)phenoxy)propanoate (**QPP-7**): white solid, yield 87.5%, m.p. 88–91 °C; $^1$H NMR (500 MHz, CDCl$_3$) δ: 7.86–7.80 (m, 1H), 7.38–7.28 (m, 2H), 7.19–7.11 (m, 2H), 6.99–6.89 (m, 2H), 4.76 (q, *J* = 6.8 Hz, 1H), 4.28–4.23 (m, 2H), 3.69 (s, 3H), 1.65 (d, *J* = 6.8 Hz, 3H), 1.28 (t, *J* = 7.1 Hz, 3H); $^{13}$C NMR (125 MHz, CDCl$_3$) δ: 172.0, 162.4 (d, *J* = 3.6 Hz), 159.7 (d, *J* = 245.5 Hz), 155.5, 152.2 (d, *J* = 1.3 Hz), 145.8, 143.1 (d, *J* = 1.5 Hz), 128.1 (d, *J* = 8.1 Hz), 122.8 (d, *J* = 24.1 Hz), 122.7, 119.8 (d, *J* = 8.5 Hz), 116.7, 116.1, 115.9, 111.9 (d, *J* = 23.7 Hz), 73.2, 61.4, 28.9, 18.6, 14.2; HRMS, m/z calcd. for $C_{20}H_{20}FN_2O_5^+$ [M + H]$^+$ 387.1351, found 387.1533.

(R)-ethyl 2-(4-((7-fluoro-3-methyl-4-oxo-3,4-dihydroquinazolin-2-yl)oxy)phenoxy)propanoate (**QPP-8**): white solid, yield 88.2%, m.p. 61–64 °C; $^1$H NMR (500 MHz, CDCl$_3$) δ: 8.19 (dd, *J* = 8.8, 6.2 Hz, 1H), 7.22–7.11 (m, 2H), 7.05–6.97 (m, 2H), 6.97–6.92 (m, 2H), 4.77 (q, *J* = 6.8 Hz, 1H), 4.25 (qd, *J* = 7.1, 3.2 Hz, 2H), 3.67 (s, 3H), 1.65 (d, *J* = 6.8 Hz, 3H), 1.28 (t, *J* = 7.1 Hz, 3H); $^{13}$C NMR (125 MHz, CDCl$_3$) δ: 172.0, 166.6 (d, *J* = 253.1 Hz), 162.2, 155.5, 153.5, 148.8 (d, *J* = 13.8 Hz), 145.7, 129.7 (d, *J* = 10.9 Hz), 122.7, 115.9, 115.6, 113.6 (d, *J* = 23.4 Hz), 111.4 (d, *J* = 22.5 Hz), 73.2, 61.4, 28.8, 18.6, 14.2; HRMS, m/z calcd. for $C_{20}H_{20}FN_2O_5^+$ [M + H]$^+$ 387.1351, found 387.1365.

(R)-ethyl 2-(4-((8-fluoro-3-methyl-4-oxo-3,4-dihydroquinazolin-2-yl)oxy)phenoxy)propanoate (**QPP-9**): white solid, yield 80.0%, m.p. 67–70 °C; $^1$H NMR (500 MHz, CDCl$_3$) δ: 7.99 (d, *J* = 8.0 Hz, 1H), 7.36–7.30 (m, 1H), 7.25 (dd, *J* = 8.0, 3.4 Hz, 1H), 7.23–7.19 (m, 2H), 7.02–6.87 (m, 2H), 4.77 (q, *J* = 6.8 Hz, 1H), 4.30–4.15 (m, 2H), 3.70 (s, 3H), 1.65 (d, *J* = 6.8 Hz, 3H), 1.27 (t, *J* = 7.1 Hz, 3H); $^{13}$C NMR (125 MHz, CDCl$_3$) δ: 172.1, 162.2, 162.1, 154.9 (d, *J* = 273.1 Hz), 155.5, 152.8, 145.8, 136.1 (d, *J* = 12.3 Hz), 124.7 (d, *J* = 7.2 Hz), 122.5, 120.9 (d, *J* = 1.6 Hz), 119.8 (d, *J* = 18.5 Hz), 115.9, 73.3, 61.4, 28.9, 18.6, 14.1; HRMS, m/z calcd. for $C_{20}H_{20}FN_2O_5^+$ [M + H]$^+$ 387.1351, found 387.1361.

(R)-ethyl 2-(4-((5-chloro-3-methyl-4-oxo-3,4-dihydroquinazolin-2-yl)oxy)phenoxy)propanoate (**QPP-10**): white solid, yield 71.5%, m.p. 90–92 °C; $^1$H NMR (500 MHz, CDCl$_3$) δ: 8.12–8.10 (m, 1H), 7.68 (dd, *J* = 7.7, 1.3 Hz, 1H), 7.31–7.26 (m, 2H), 7.25–7.20 (m, 1H), 6.97–6.91 (m, 2H), 4.78 (q, *J* = 6.8 Hz, 1H), 4.28–4.14 (m, 2H), 3.69 (s, 3H), 1.65 (d, *J* = 6.8 Hz, 3H), 1.26 (t, *J* = 7.1 Hz, 3H); $^{13}$C NMR (125 MHz, CDCl$_3$) δ: 172.1, 162.6, 155.3, 152.7, 145.9, 143.5, 134.5, 130.4, 125.8, 124.9, 122.5, 120.4, 115.8, 73.2, 61.4, 28.9, 18.6, 14.1; HRMS, m/z calcd. for $C_{20}H_{20}ClN_2O_5^+$ [M + H]$^+$ 403.1055, found 403.1060.

(R)-ethyl 2-(4-((6-chloro-3-methyl-4-oxo-3,4-dihydroquinazolin-2-yl)oxy)phenoxy)propanoate (**QPP-11**): white solid, yield 91.7%, m.p. 100–102 °C; $^1$H NMR (500 MHz, CDCl$_3$) δ: 8.16 (d, *J* = 2.4 Hz, 1H), 7.52 (dd, *J* = 8.7, 2.5 Hz, 1H), 7.28 (d, *J* = 8.7 Hz, 1H), 7.17–7.12 (m, 2H), 6.97–6.92 (m, 2H), 4.76 (q, *J* = 6.8 Hz, 1H), 4.25 (qd, *J* = 7.1, 2.6 Hz, 2H), 3.69 (s, 3H), 1.65 (d, *J* = 6.8 Hz, 3H), 1.28 (t, *J* = 7.1 Hz, 3H); $^{13}$C NMR (125 MHz, CDCl$_3$) δ: 172.0, 162.0, 155.5, 152.8, 145.7, 145.2, 134.6, 130.5, 127.6, 126.4, 122.7, 119.9, 115.9, 73.2, 61.4, 28.9, 18.6, 14.2; HRMS, m/z calcd. for $C_{20}H_{20}ClN_2O_5^+$ [M + H]$^+$ 403.1055, found 403.1066.

(R)-ethyl 2-(4-((7-chloro-3-methyl-4-oxo-3,4-dihydroquinazolin-2-yl)oxy)phenoxy)propanoate (**QPP-12**): white solid, yield 80.6%, m.p. 77–80 °C; $^1$H NMR (500 MHz, CDCl$_3$) δ: 7.27 (t, *J* = 8.0 Hz, 1H), 7.17–7.13 (m, 1H), 7.10–7.02 (m, 3H), 6.83 (dd, *J* = 9.8, 2.9 Hz, 2H), 4.65 (q, *J* = 6.8 Hz, 1H), 4.16–4.10 (m, 2H), 3.52 (s, 3H), 1.53 (d, *J* = 6.8 Hz, 3H), 1.16 (t, *J* = 7.2 Hz, 3H); $^{13}$C NMR (125 MHz, CDCl$_3$) δ: 170.9, 159.8, 154.4, 151.7, 148.1, 144.6, 133.1, 132.4, 126.6, 124.2, 121.7, 114.9, 114.8, 72.1, 60.3, 27.8, 17.5, 13.1; HRMS, m/z calcd. for $C_{20}H_{20}ClN_2O_5^+$ [M + H]$^+$ 403.1055, found 403.1064.

(R)-ethyl 2-(4-((8-chloro-3-methyl-4-oxo-3,4-dihydroquinazolin-2-yl)oxy)phenoxy)propanoate (**QPP-13**): white solid, yield 85.8%, m.p. 97–98 °C; [1]H NMR (500 MHz, CDCl$_3$) δ: 7.27 (t, $J$ = 8.0 Hz, 1H), 7.17–7.12 (m, 1H), 7.10–7.02 (m, 3H), 6.83 (dd, $J$ = 9.8, 2.9 Hz, 2H), 4.65 (q, $J$ = 6.8 Hz, 1H), 4.16–4.10 (m, 2H), 3.52 (s, 3H), 1.53 (d, $J$ = 6.8 Hz, 3H), 1.16 (t, $J$ = 7.2 Hz, 3H); [13]C NMR (125 MHz, CDCl$_3$) δ: 170.9, 159.8, 154.4, 151.7, 148.1, 144.6, 133.1, 132.4, 126.6, 124.2, 121.7, 114.9, 114.8, 72.1, 60.3, 27.8, 17.5, 13.1; HRMS, m/z calcd. for $C_{20}H_{20}ClN_2O_5^+$ [M + H]$^+$ 403.1055, found 403.1067.

(R)-ethyl 2-(4-((3,6,7-trimethyl-4-oxo-3,4-dihydroquinazolin-2-yl)oxy)phenoxy)propanoate (**QPP-14**): white solid, yield 89.8%, m.p. 124–125 °C; [1]H NMR (500 MHz, CDCl$_3$) δ: 7.93 (s, 1H), 7.19–7.09 (m, 3H), 6.94 (d, $J$ = 8.6 Hz, 2H), 4.75 (q, $J$ = 6.7 Hz, 1H), 4.31–4.17 (m, 2H), 3.67 (dd, $J$ = 3.5, 2.5 Hz, 3H), 2.31 (d, $J$ = 17.4 Hz, 3H), 1.65 (d, $J$ = 6.8 Hz, 3H), 1.28 (t, $J$ = 7.1 Hz, 3H); [13]C NMR (125 MHz, CDCl$_3$) δ: 172.1, 162.9, 155.3, 152.3, 146.0, 144.8, 144.5, 134.2, 134.2, 126.8, 126.3, 122.7, 116.6, 115.9, 73.2, 61.4, 28.7, 20.2, 19.5, 18.6, 14.2; HRMS, m/z calcd. for $C_{22}H_{25}N_2O_5^+$ [M + H]$^+$ 397.1758, found 397.1765.

(R)-ethyl 2-(4-((6,7-difluoro-3-methyl-4-oxo-3,4-dihydroquinazolin-2-yl)oxy)phenoxy)-propanoate (**QPP-15**): white solid, yield 84.6%, m.p. 76–79 °C; [1]H NMR (500 MHz, CDCl$_3$) δ: 7.81 (dd, $J$ = 9.8, 8.9 Hz, 1H), 7.07–7.02 (m, 2H), 7.02–6.97 (m, 1H), 6.88–6.80 (m, 2H), 4.67 (q, $J$ = 6.8 Hz, 1H), 4.19–4.12 (m, 2H), 3.57 (s, 3H), 1.55 (d, $J$ = 6.8 Hz, 3H), 1.19 (t, $J$ = 7.1 Hz, 3H); [13]C NMR (125 MHz, CDCl$_3$) δ: 171.9, 161.5 (d, $J$ = 2.7 Hz), 155.6, 154.84 (dd, $J$ = 256.25 Hz,14.5 Hz), 153.8 (dd, $J$ = 246.25 Hz, 14.6 Hz), 153.2, 145.6, 144.29 (dd, $J$ = 11.8, 1.6 Hz), 122.6, 115.9, 115.43 (d, $J$ = 4.7 Hz), 114.26 (dd, $J$ = 19.2, 1.5 Hz), 113.72 (d, $J$ = 18.3 Hz), 73.1, 61.4, 28.9, 18.5, 14.1; HRMS, m/z calcd. for $C_{20}H_{19}F_2N_2O_5^+$ [M + H]$^+$ 405.1257, found 405.1266.

(R)-ethyl 2-(4-((3-ethyl-6-fluoro-4-oxo-3,4-dihydroquinazolin-2-yl)oxy)phenoxy)propanoate (**QPP-16**): white solid, yield 91.4%, m.p. 124–127 °C; [1]H NMR (500 MHz, CDCl$_3$) δ: 7.90–7.77 (m, 1H), 7.35–7.29 (m, 2H), 7.17–7.12 (m, 2H), 6.97–6.90 (m, 2H), 4.76 (q, $J$ = 6.8 Hz, 1H), 4.33 (q, $J$ = 7.1 Hz, 2H), 4.29–4.21 (m, 2H), 1.65 (d, $J$ = 6.8 Hz, 3H), 1.42 (t, $J$ = 7.1 Hz, 3H), 1.28 (t, $J$ = 7.1 Hz, 3H); [13]C NMR (125 MHz, CDCl$_3$) δ: 172.0, 161.9 (d, $J$ = 3.4 Hz), 159.7 (d, $J$ = 245.3 Hz), 155.4, 152.1, 145.8, 143.2, 128.1 (d, $J$ = 8.0 Hz), 122.8, 122.6, 120.2 (d, $J$ = 8.5 Hz), 115.9, 111.9 (d, $J$ = 23.5 Hz), 73.2, 61.4, 37.6, 18.6, 14.2, 13.8; HRMS, m/z calcd. for $C_{21}H_{22}FN_2O_5^+$ [M + H]$^+$ 401.1507, found 405.1514.

(R)-ethyl 2-(4-((6-fluoro-4-oxo-3-propyl-3,4-dihydroquinazolin-2-yl)oxy)phenoxy)propanoate (**QPP-17**): white solid, yield 79.7%, m.p. 122–123 °C; [1]H NMR (500 MHz, CDCl$_3$) δ: 7.83 (d, $J$ = 8.5 Hz, 1H), 7.32 (d, $J$ = 5.5 Hz, 2H), 7.14 (d, $J$ = 8.9 Hz, 2H), 6.94 (d, $J$ = 8.8 Hz, 2H), 4.76 (q, $J$ = 6.8 Hz, 1H), 4.28–4.21 (m, 4H), 1.97–1.78 (m, 2H), 1.65 (d, $J$ = 6.8 Hz, 3H), 1.29 (t, $J$ = 7.1 Hz, 3H), 1.03 (t, $J$ = 7.4 Hz, 3H); [13]C NMR (125 MHz, CDCl$_3$) δ: 172.0, 162.1 (d, $J$ = 3.1 Hz), 159.7 (d, $J$ = 245.5 Hz), 155.4, 152.2, 145.9, 143.2, 128.1 (d, $J$ = 8.0 Hz), 122.8, 122.6, 120.1 (d, $J$ = 8.4 Hz), 115.9, 111.9 (d, $J$ = 23.7 Hz), 73.2, 61.4, 43.9, 21.9, 18.6, 14.2, 11.4; HRMS, m/z calcd. for $C_{22}H_{24}FN_2O_5^+$ [M + H]$^+$ 415.1664, found 415.1675.

(R)-ethyl 2-(4-((3-butyl-6-fluoro-4-oxo-3,4-dihydroquinazolin-2-yl)oxy)phenoxy)propanoate (**QPP-18**): white solid, yield 75.1%, m.p. 108–109 °C; [1]H NMR (500 MHz, CDCl$_3$) δ: 7.89–7.71 (m, 1H), 7.35–7.31 (m, 2H), 7.16–7.11 (m, 2H), 6.97–6.92 (m, 2H), 4.76 (q, $J$ = 6.8 Hz, 1H), 4.34–4.18 (m, 4H), 1.89–1.74 (m, 2H), 1.65 (d, $J$ = 6.8 Hz, 3H), 1.55–1.38 (m, 2H), 1.29 (t, $J$ = 7.1 Hz, 3H), 0.99 (t, $J$ = 7.4 Hz, 3H); [13]C NMR (125 MHz, CDCl$_3$) δ: 172.0, 162.1 (d, $J$ = 3.3 Hz), 159.7 (d, $J$ = 245.3 Hz), 155.4, 152.2, 145.9, 143.2, 128.1 (d, $J$ = 7.8 Hz), 122.8, 122.6, 120.1 (d, $J$ = 8.4 Hz), 115.9, 111.9 (d, $J$ = 23.5 Hz), 73.2, 61.4, 42.3, 30.6, 20.2, 18.6, 14.2, 13.8; HRMS, m/z calcd. for $C_{23}H_{26}FN_2O_5^+$ [M + H]$^+$ 429.1820, found 429.1830.

(R)-ethyl 2-(4-((6-fluoro-3-isobutyl-4-oxo-3,4-dihydroquinazolin-2-yl)oxy)phenoxy)propanoate (**QPP-19**): white solid, yield 85.8%, m.p. 100–103 °C; [1]H NMR (500 MHz, CDCl$_3$) δ: 7.85–7.80 (m, 1H), 7.37–7.30 (m, 2H), 7.17–7.10 (m, 2H), 6.98–6.92 (m, 2H), 4.76 (q, $J$ = 6.8 Hz, 1H), 4.31–4.20 (m, 2H), 4.10 (d, $J$ = 7.5 Hz, 2H), 2.29 (dp, $J$ = 14.0, 7.0 Hz, 1H), 1.65 (d, $J$ = 6.8 Hz, 3H), 1.29 (t, $J$ = 7.1 Hz, 3H), 1.02 (s, 3H), 1.01 (s, 3H); [13]C NMR (125 MHz, CDCl$_3$) δ: 172.0, 162.3 (d, $J$ = 3.2 Hz), 159.7 (d, $J$ = 245.4 Hz), 155.4, 152.4, 145.8, 143.2, 128.1 (d, $J$ = 7.7 Hz), 122.8, 122.6 (d, $J$ = 2.6 Hz), 120.1 (d, $J$ = 8.4 Hz), 115.9, 112.0 (d, $J$ = 23.8 Hz), 73.2, 61.4,

49.2, 29.7, 27.8, 20.2, 18.6, 14.2; HRMS, m/z calcd. for $C_{23}H_{26}FN_2O_5^+$ [M + H]$^+$ 429.1820, found 429.1830.

(R)-ethyl 2-(4-((6-fluoro-4-oxo-3-phenyl-3,4-dihydroquinazolin-2-yl)oxy)phenoxy)propanoate (**QPP-20**): white solid, yield 69.7%, m.p. 135–136 °C; $^1$H NMR (500 MHz, CDCl$_3$) δ: 7.86 (dd, *J* = 8.4, 2.8 Hz, 1H), 7.54 (t, *J* = 7.5 Hz, 2H), 7.51–7.46 (m, 1H), 7.43–7.36 (m, 4H), 7.05 (d, *J* = 9.0 Hz, 2H), 6.87 (d, *J* = 9.0 Hz, 2H), 4.72 (q, *J* = 6.8 Hz, 1H), 4.27–4.17 (m, 2H), 1.62 (d, *J* = 6.8 Hz, 3H), 1.25 (t, *J* = 7.1 Hz, 3H); $^{13}$C NMR (125 MHz, CDCl$_3$) δ: 172.0, 162.2 (d, *J* = 3.5 Hz), 159.9 (d, *J* = 246.2 Hz), 155.4, 151.6, 145.8, 143.3, 134.9, 129.5, 129.1, 128.3 (d, *J* = 7.8 Hz), 128.1, 123.1 (d, *J* = 24.0 Hz), 122.5, 120.5 (d, *J* = 8.8 Hz), 115.8, 112.3 (d, *J* = 23.7 Hz), 73.1, 61.4, 18.6, 14.1; HRMS, m/z calcd. for $C_{25}H_{22}FN_2O_5^+$ [M + H]$^+$ 449.1507, found 449.1516.

(R)-ethyl 2-(4-((6-fluoro-4-oxo-3-(*o*-tolyl)-3,4-dihydroquinazolin-2-yl)oxy)phenoxy)propanoate (**QPP-21**): white solid, yield 69.5%, m.p. 47–49 °C; $^1$H NMR (500 MHz, CDCl$_3$) δ: 7.88 (dd, *J* = 8.4, 2.8 Hz, 1H), 7.47–7.33 (m, 5H), 7.25 (t, *J* = 3.6 Hz, 1H), 7.06–7.01 (m, 2H), 6.90–6.84 (m, 2H), 4.72 (q, *J* = 6.8 Hz, 1H), 4.22 (q, *J* = 7.1 Hz, 2H), 2.23 (s, 3H), 1.62 (d, *J* = 6.8 Hz, 3H), 1.25 (t, *J* = 7.1, 3H); $^{13}$C NMR (125 MHz, CDCl$_3$) δ: 171.9, 161.8 (d, *J* = 3.5 Hz), 159.9 (d, *J* = 246.2 Hz), 155.4, 151.7, 145.8, 143.6, 135.5, 134.2, 131.2, 129.5, 127.3, 123.1 (d, *J* = 24.0 Hz), 122.5, 120.5 (d, *J* = 8.6 Hz), 115.8, 112.4 (d, *J* = 23.8 Hz), 73.1, 61.4, 18.6, 17.6, 14.2; HRMS, m/z calcd. for $C_{26}H_{24}FN_2O_5^+$ [M + H]$^+$ 463.1664, found 463.1672.

(R)-ethyl 2-(4-((6-fluoro-4-oxo-3-(*m*-tolyl)-3,4-dihydroquinazolin-2-yl)oxy)phenoxy)propanoate (**QPP-22**): white solid, yield 84.3%, m.p. 119–122 °C; $^1$H NMR (500 MHz, CDCl$_3$) δ: 7.86 (dd, *J* = 8.4, 2.8 Hz, 1H), 7.39 (ddt, *J* = 11.8, 8.9, 5.4 Hz, 3H), 7.29 (d, *J* = 7.7 Hz, 1H), 7.16 (d, *J* = 8.5 Hz, 2H), 7.08–7.02 (m, 2H), 6.91–6.85 (m, 2H), 4.72 (q, *J* = 6.8 Hz, 1H), 4.27–4.13 (m, 2H), 2.43 (s, 3H), 1.62 (d, *J* = 6.8 Hz, 3H), 1.26 (t, *J* = 7.1 Hz, 3H); $^{13}$C NMR (125 MHz, CDCl$_3$) δ: 172.0, 162.3 (d, *J* = 3.5 Hz), 159.9 (d, *J* = 246.0 Hz), 155.4, 151.7, 145.9, 143.4, 139.6, 134.9, 129.9, 129.3, 128.6, 128.3 (d, *J* = 8.0 Hz), 124.9, 123.1 (d, *J* = 23.9 Hz), 122.6, 120.6 (d, *J* = 8.8 Hz), 115.8, 112.3 (d, *J* = 23.7 Hz), 73.2, 61.4, 29.7, 21.4, 18.6, 14.2; HRMS, m/z calcd. for $C_{26}H_{24}FN_2O_5^+$ [M + H]$^+$ 463.1664, found 463.1669.

(R)-ethyl 2-(4-((6-fluoro-4-oxo-3-(*p*-tolyl)-3,4-dihydroquinazolin-2-yl)oxy)phenoxy)propanoate (**QPP-23**): white solid, yield 68.0%, m.p. 151–152 °C; $^1$H NMR (500 MHz, CDCl$_3$) δ: 7.86 (dd, *J* = 8.4, 2.8 Hz, 1H), 7.44–7.32 (m, 4H), 7.24 (d, *J* = 8.3 Hz, 2H), 7.07–7.02 (m, 2H), 6.90–6.85 (m, 2H), 4.72 (q, *J* = 6.8 Hz, 1H), 4.25–4.19 (m, 2H), 2.43 (s, 3H), 1.62 (d, *J* = 6.8 Hz, 3H), 1.25 (t, *J* = 7.1 Hz, 3H); $^{13}$C NMR (125 MHz, CDCl$_3$) δ: 172.0, 162.3 (d, *J* = 3.5 Hz), 159.9 (d, *J* = 245.7 Hz), 155.3, 151.8, 145.9, 143.3, 139.1, 132.3, 130.2, 128.3 (d, *J* = 8.0 Hz), 127.7, 123.1 (d, *J* = 24.0 Hz), 122.5, 120.6 (d, *J* = 8.6 Hz), 115.8, 112.4 (d, *J* = 23.8 Hz), 73.2, 61.4, 21.3, 18.6, 14.2; HRMS, m/z calcd. for $C_{26}H_{24}FN_2O_5^+$ [M + H]$^+$ 463.1664, found 463.1670.

(R)-ethyl 2-(4-((3-(2-chlorophenyl)-6-fluoro-4-oxo-3,4-dihydroquinazolin-2-yl)oxy)phenoxy)propanoate (**QPP-24**): white solid, yield 74.3%, m.p. 51–54 °C; $^1$H NMR (500 MHz, CDCl$_3$) δ: 7.88 (dd, *J* = 8.3, 2.8 Hz, 1H), 7.64–7.57 (m, 1H), 7.50–7.36 (m, 5H), 7.10–7.05 (m, 2H), 6.93–6.84 (m, 2H), 4.72 (q, *J* = 6.8 Hz, 1H), 4.22 (q, *J* = 7.1 Hz, 2H), 1.62 (d, *J* = 6.8 Hz, 3H), 1.26 (t, *J* = 7.1 Hz, 3H); $^{13}$C NMR (125 MHz, CDCl$_3$) δ: 171.9, 161.5 (d, *J* = 3.0 Hz), 159.9 (d, *J* = 246.3 Hz), 155.5, 151.2, 145.7, 143.5, 132.9, 132.4, 130.5 (d, *J* = 29.2 Hz), 130.1, 128.4 (d, *J* = 8.0 Hz), 128.0, 123.3 (d, *J* = 23.9 Hz), 122.6, 120.3 (d, *J* = 8.8 Hz), 115.8, 112.5 (d, *J* = 23.7 Hz), 73.1, 61.4, 18.6, 14.2; HRMS, m/z calcd. for $C_{25}H_{21}ClFN_2O_5^+$ [M + H]$^+$ 483.1118, found 483.1122.

(R)-ethyl 2-(4-((3-(3-chlorophenyl)-6-fluoro-4-oxo-3,4-dihydroquinazolin-2-yl)oxy)phenoxy)propanoate (**QPP-25**): white solid, yield 82.0%, m.p. 113–116 °C; $^1$H NMR (500 MHz, CDCl$_3$) δ: 7.85 (dd, *J* = 8.3, 2.6 Hz, 1H), 7.51–7.45 (m, 2H), 7.42–7.38 (m, 3H), 7.29–7.26 (m, 1H), 7.08–7.03 (m, 2H), 6.92–6.85 (m, 2H), 4.73 (q, *J* = 6.8 Hz, 1H), 4.22 (qd, *J* = 7.1, 0.7 Hz, 2H), 1.63 (d, *J* = 6.8 Hz, 3H), 1.26 (t, *J* = 7.1 Hz, 3H); $^{13}$C NMR (125 MHz, CDCl$_3$) δ: 171.9, 161.9 (d, *J* = 3.5 Hz), 160.0 (d, *J* = 246.5 Hz), 155.5, 151.1, 145.7, 143.2, 135.9, 135.1, 130.5, 129.5, 128.6, 128.4 (d, *J* = 8.1 Hz), 126.6, 123.3 (d, *J* = 24.0 Hz), 122.5, 120.4 (d, *J* = 8.7 Hz), 115.8, 112.4 (d, *J* = 23.9 Hz), 73.1, 61.4, 18.6, 14.2; HRMS, m/z calcd. for $C_{25}H_{21}ClFN_2O_5^+$ [M + H]$^+$ 483.1118, found 483.1124.

(R)-ethyl 2-(4-((3-(4-chlorophenyl)-6-fluoro-4-oxo-3,4-dihydroquinazolin-2-yl)oxy)phenoxy)propanoate (**QPP-26**): white solid, yield 78.1%, m.p. 146–148 °C; $^1$H NMR (500 MHz, CDCl$_3$) $\delta$: 7.85 (dd, $J$ = 8.3, 2.5 Hz, 1H), 7.52 (d, $J$ = 8.4 Hz, 2H), 7.45–7.36 (m, 2H), 7.31 (d, $J$ = 8.5 Hz, 2H), 7.05 (d, $J$ = 8.5 Hz, 2H), 6.88 (d, $J$ = 8.3 Hz, 2H), 4.72 (q, $J$ = 6.7 Hz, 1H), 4.22 (q, $J$ = 7.0 Hz, 2H), 1.63 (d, $J$ = 6.8 Hz, 3H), 1.26 (t, $J$ = 7.1 Hz, 3H); $^{13}$C NMR (125 MHz, CDCl$_3$) $\delta$: 171.9, 162.0 (d, $J$ = 3.1 Hz), 160.0 (d, $J$ = 246.2 Hz), 155.5, 151.2, 145.7, 143.2, 135.2, 133.4, 128.4 (d, $J$ = 8.0 Hz), 123.3 (d, $J$ = 24.0 Hz), 122.4, 115.9, 112.4 (d, $J$ = 23.9 Hz), 73.2, 61.4, 18.6, 14.2; HRMS, m/z calcd. for C$_{25}$H$_{21}$ClFN$_2$O$_5^+$ [M + H]$^+$ 483.1118, found 483.1123.

(R)-ethyl 2-(4-((6-fluoro-4-oxo-3-(2-(trifluoromethyl)phenyl)-3,4-dihydroquinazolin-2-yl)oxy)phenoxy)propanoate (**QPP-27**): white solid, yield 96.9%, m.p. 45–48 °C; $^1$H NMR (500 MHz, CDCl$_3$) $\delta$: 7.77 (d, $J$ = 8.1 Hz, 2H), 7.66 (t, $J$ = 7.7 Hz, 1H), 7.54 (t, $J$ = 7.7 Hz, 1H), 7.38 (d, $J$ = 7.9 Hz, 1H), 7.35–7.26 (m, 2H), 6.93 (d, $J$ = 8.8 Hz, 2H), 6.78 (d, $J$ = 8.8 Hz, 2H), 4.62 (q, $J$ = 6.8 Hz, 1H), 4.12 (q, $J$ = 7.1 Hz, 2H), 1.52 (d, $J$ = 6.8 Hz, 3H), 1.16 (t, $J$ = 7.1 Hz, 3H); $^{13}$C NMR (125 MHz, CDCl$_3$) $\delta$: 171.9, 161.9 (d, $J$ = 3.3 Hz), 159.9 (d, $J$ = 246.2 Hz), 155.5, 151.3, 145.6, 143.5, 133.4, 133.2 (d, $J$ = 1.5 Hz), 130.8, 129.9, 128.5 (d, $J$ = 7.9 Hz), 127.9 (q, $J$ = 31.2 Hz), 127.7 (q, $J$ = 4.2 Hz), 123.3 (d, $J$ = 23.9 Hz), 123.1 (q, $J$ = 216.7 Hz), 122.5, 120.1 (d, $J$ = 8.8 Hz), 115.8, 112.4 (d, $J$ = 23.9 Hz), 73.1, 61.4, 18.6, 14.1; HRMS, m/z calcd. for C$_{26}$H$_{21}$F$_4$N$_2$O$_5^+$ [M + H]$^+$ 517.1381, found 517.1386.

(R)-ethyl 2-(4-((6-fluoro-4-oxo-3-(3-(trifluoromethyl)phenyl)-3,4-dihydroquinazolin-2-yl)oxy)phenoxy)propanoate (**QPP-28**): white solid, yield 85.0%, m.p. 39–42 °C; $^1$H NMR (500 MHz, CDCl$_3$) $\delta$: 7.86 (dd, $J$ = 8.2, 2.6 Hz, 1H), 7.76 (d, $J$ = 7.9 Hz, 1H), 7.72–7.66 (m, 2H), 7.58 (d, $J$ = 7.9 Hz, 1H), 7.45–7.38 (m, 2H), 7.08–7.03 (m, 2H), 6.91–6.86 (m, 2H), 4.72 (q, $J$ = 6.8 Hz, 1H), 4.22 (q, $J$ = 7.1 Hz, 2H), 1.63 (d, $J$ = 6.8 Hz, 3H), 1.26 (t, $J$ = 7.1 Hz, 3H); $^{13}$C NMR (125 MHz, CDCl$_3$) $\delta$: 171.9, 161.9 (d, $J$ = 3.2 Hz), 160.1 (d, $J$ = 246.6 Hz), 155.5, 150.9, 145.6, 143.2, 135.5, 132.2 (q, $J$ = 33.1 Hz), 131.8, 130.2, 128.5 (d, $J$ = 7.8 Hz), 126.1 (q, $J$ = 3.2 Hz), 125.5 (q, $J$ = 3.6 Hz), 123.5 (d, $J$ = 23.9 Hz), 122.4, 120.3 (d, $J$ = 8.4 Hz), 115.8, 112.4 (d, $J$ = 23.9 Hz), 73.1, 61.4, 18.6, 14.2; HRMS, m/z calcd. for C$_{26}$H$_{21}$F$_4$N$_2$O$_5^+$ [M + H]$^+$ 517.1381, found 517.1387.

(R)-ethyl 2-(4-((6-fluoro-4-oxo-3-(4-(trifluoromethyl)phenyl)-3,4-dihydroquinazolin-2-yl)oxy)phenoxy)propanoate (**QPP-29**): white solid, yield 77.0%, m.p. 146–147 °C; $^1$H NMR (500 MHz, CDCl$_3$) $\delta$: 7.86 (dd, $J$ = 8.2, 2.6 Hz, 1H), 7.82 (d, $J$ = 8.4 Hz, 2H), 7.52 (d, $J$ = 8.2 Hz, 2H), 7.46–7.37 (m, 2H), 7.07–7.02 (m, 2H), 6.91–6.85 (m, 2H), 4.72 (q, $J$ = 6.8 Hz, 1H), 4.22 (qd, $J$ = 7.1, 0.9 Hz, 2H), 1.63 (d, $J$ = 6.8 Hz, 3H), 1.26 (t, $J$ = 7.1 Hz, 3H); $^{13}$C NMR (125 MHz, CDCl$_3$) $\delta$: 171.9, 161.9 (d, $J$ = 3.4 Hz), 160.1 (d, $J$ = 246.8 Hz), 155.5, 150.9, 145.6, 143.2, 138.1, 131.4 (q, $J$ = 32.9 Hz), 128.9, 128.5 (d, $J$ = 7.9 Hz), 126.7 (q, $J$ = 7.3 Hz), 123.5 (d, $J$ = 23.9 Hz), 122.4, 120.3 (d, $J$ = 8.3 Hz), 112.4 (d, $J$ = 23.8 Hz), 73.1, 61.4, 18.6, 14.2; HRMS, m/z calcd. for C$_{26}$H$_{21}$F$_4$N$_2$O$_5^+$ [M + H]$^+$ 517.1381, found 517.1387.

### 2.3. X-ray Diffraction Analysis of Target Compound QPP-7

The single crystal of **QPP-7** was slowly cultivated from a mixture of dichloromethane and ethanol (1/1 by volume). Crystallographic data of compound **QPP-7** had been deposited with the Cambridge Crystallographic Data Centre as supplementary publications with the deposition number 2183936. The detail data can be acquired free of charge from http://www.ccdc.cam.ac.uk/ (15 July 2022).

### 2.4. Evaluation of Herbicidal Activity

Herbicidal activity was evaluated based on the reported methods [34]. Commercial herbicide quizalofop-P-ethyl (QZ) was selected as the positive control. The preliminary in vitro herbicidal activity of target compounds **QPP-1** to **QPP-15** was determined with *Brassica campestris* root test and *Echinochloa crusgalli* cup test at a dosage of 10 μg/mL. Further herbicidal activity of target compounds **QPP-1** to **QPP-29** against two dicotyledonous species *Brassica campestris* and *Amaranthus retroflexus*, and two monocotyledonous species *Echinochloa crusgalli* and *Digitaria sanguinalis* was tested in a greenhouse. Briefly, the target compounds were dissolved in 100 μL of *N*, *N*-dimethylformamide with the addition

of a little Tween 80 and then were sprayed using a laboratory belt sprayer delivering a 750 L/ha spray volume. The dosage (activity ingredient) for each compound corresponded to 1500 g/ha. Compounds were sprayed immediately after seed planting (preemergence treatment) or after the expansion of the first true leaf (postemergence treatment). The mixture of same amount of water, *N, N*-dimethylformamide, and Tween 80 was sprayed as the control. The fresh weight of the above ground tissues was measured 14 days after treatment. The inhibition percent was used to describe the control efficiency of the compounds. The data represented the percent displaying herbicidal damage as compared to the control, where complete control of the target is 100 and no control is 0. Compounds **QPP-3**, **QPP-7**, **QPP-11** were selected to study the herbicidal activity at 750 g/ha, 375 g/ha and 187.5 g/ha, respectively. Compound **QPP-7** was selected to study the herbicidal spectrum. All of the bioassays were tested for three parallel experiments. Details of the experimental procedure is given in the Supplementary Materials section.

### 2.5. Crop Selectivity

Based on the reported methods, [34] the crop selectivity of target compound **QPP-7** was evaluated with three replicates per treatment. Six representative crops, namely, *Oryza sativa*, *Zea mays*, *Triticum aestivum*, *Gossypium spp*, *Glycine max* and *Arachis hypogaea*, were selected for crop selectivity studies in the greenhouse. The procedure is given in the Supplementary Materials section.

### 2.6. Molecular Docking Study

As we known, the APP herbicides as predrug, played major role in plants through hydrolysis into acids [39]. Therefore, the free acid of compound **QPP-7** and QZ were dock into the binding site of ACCase by using the molecular docking module (CDOCKER) in the DS software (Discovery Studio 2020, Dassault Systemes, France). The crystal structure of ACCase enzyme was extracted from crystal structure PDB code 1UYR, and $H_2O$ and all binding ligands were deleted. The structures of small molecules were optimized with the ligand minimization protocol. The free acid of compound **QPP-7** and QZ were set near the original diclofop-methyl binding site (amino acid residues from 1596 to 2025) in ACCase, respectively. The combined spherical area were (x = 26.6004, y = 40.6731, z = 77.6454, Radius = 10). After calculations with the parameters set as default values, the generated conformations were clustered together and ranked by the lowest docking energy, and a cluster analysis was performed.

### 2.7. ACCase Extraction and Inhibition Activity Assay

When *E. crusgalli* was grown to the 3-leaf stage in a greenhouse, shoots were cut at the base and stored at $-80\ ^\circ C$. ACCase was extracted and partially purified using the method described by Cocker et al. [40]. The ACCase enzyme inhibition assay was performed by Shanghai Enzyme-linked Biotechnology Co., Ltd. (Shanghai, China). The ACCase enzyme was treated with the inhibitors **QPP-7** and QZ, and the enzyme activity was measured using the method of enzyme-linked immune-sorbent assay (ELISA) according to the manufacturer's instructions (Shanghai Enzyme-linked Biotechnology Co., Ltd., Shanghai, China). The inhibitor concentrations ranged from 6.25 to 100 nM, and each experiment was repeated at least three times. The absorbance [optical density (OD) value] was determined at 450 nm and used to calculate the half maximal inhibitory concentration ($IC_{50}$) values.

## 3. Results and Discussion

### 3.1. Synthetic Chemistry

As depicted in Scheme 1, the target compounds **QPP-1** to **QPP-15** were prepared via a three-step synthetic route using methyl isothiocyanate and several anthranilic acids as the starting material (**Route A**); the target compounds **QPP-16** to **QPP-29** were prepared via a three-step synthetic route using 2-amino-5-fluorobenzoic acid and several isothio-

cyanates as the starting material (**Route B**). Briefly, anthranilic acids **1a–1o** reacted with isothiocyanates in ethanol by using triethylamine as a base to provide intermediates **2a–2o** and **5a–5n**, which was then reacted with sulfuryl chloride in trichloromethane to provide intermediates **3a–3o** and **6a–6n**. Finally, intermediates **3a–3o** and **6a–6n** reacted with commercial (R)-2-(4-hydroxyphenoxy)propionic acid in acetonitrile using potassium carbonate as a base to provide target compounds **QPP-1** to **QPP-29** in 68.0% to 96.9% yields. The structures of all the target compounds were confirmed by $^1$H NMR, $^{13}$C NMR, and HRMS.

Furthermore, to demonstrate that the reaction conditions have little effect on the chiral carbon configuration of the target compounds, the enantiomeric excesses (*ee*) values of the target compound **QPP-7** and intermediate (R)-2-(4-hydroxyphenoxy)propionic acid were tested. The enantiomeric excesses were determined by HPLC analysis over a chiral column (Daicel Chiralcel OD-H, eluted with hexane-isopropyl alcohol; monitored by UV detector). The results showed that the *ee* values of (R)-2-(4-hydroxyphenoxy)propionic acid and **QPP-7** are 100% and 95.8%, respectively. In addition, the configuration of compound **QPP-7** was confirmed using the X-ray diffraction analysis (CCDC 2183936, Figure 2).

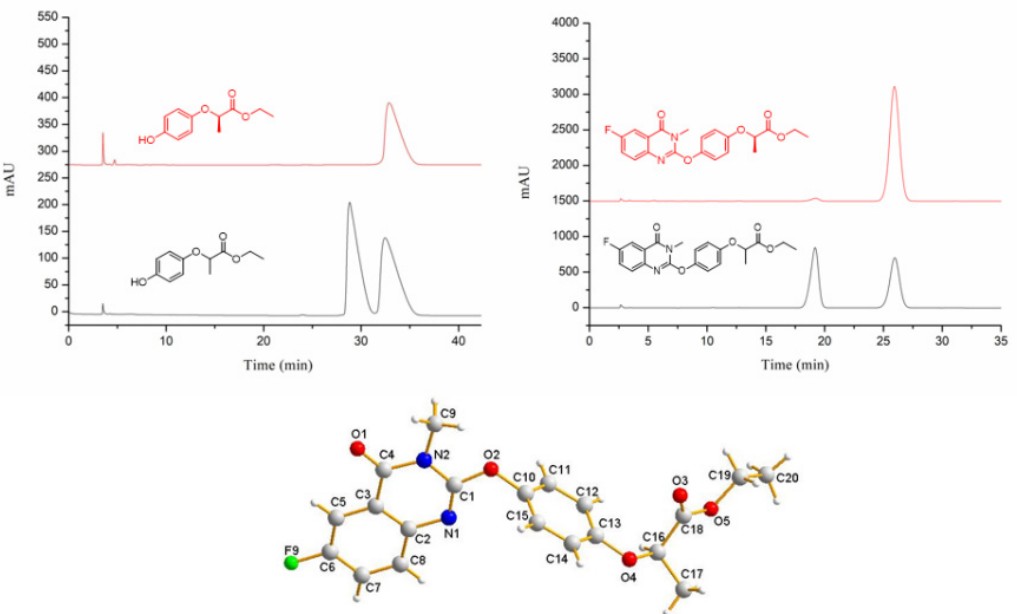

**Figure 2.** *ee* value testing and X-ray crystal structure of target compound **QPP-7**.

### 3.2. In Vitro Herbicidal Activity of Target Compounds QPP-1 to QPP-15

The in vitro herbicidal activities of the target compounds **QPP-1** to **QPP-15** were preliminarily determined by the *B. campestris* root test and *E. crusgalli* cup test at a dosage of 10 µg/mL. Commercial herbicide quizalofop-P-ethyl was selected as the positive control sample. As shown in Figure 3, some of the target compounds, such as **QPP-1** to **QPP-3**, **QPP-7** to **QPP-9**, **QPP-11**, **QPP-12**, and **QPP-15**, exhibited good herbicidal activity against the monocotyledonous plant *E. crusgalli* with >50% inhibition. Among them, compound **QPP-7** exhibited excellent herbicidal activity against *E. crusgalli* with 100% inhibition, which is equal to commercial herbicide quizalofop-P-ethyl. However, the target compounds have no inhibition against the dicotyledonous plant *B. campestris*. This finding indicated that the target compounds have stronger herbicidal activity against monocotyledonous plant.

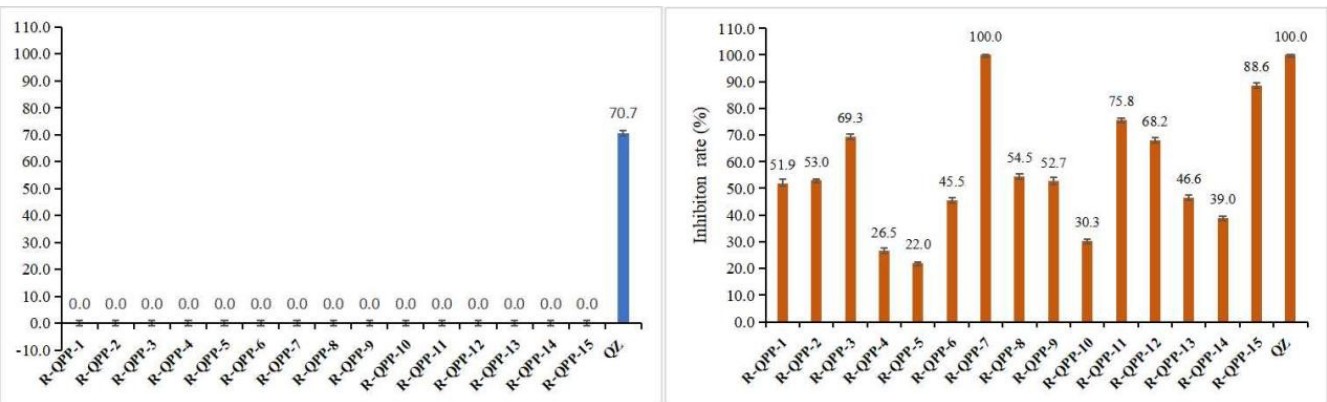

**Figure 3.** In vitro herbicidal activity of target compounds **QPP-1** to **QPP-15** at a dosage of 10 μg/mL; *B. campestris* root test (**Left**), *E. crusgalli* cup test (**Right**).

### 3.3. Herbicidal Activity of Target Compounds QPP-1 to QPP-29 in Greenhouse Tests and SAR Study

Based on the above preliminary bioassay results, the herbicidal activity of target compounds **QPP-1** to **QPP-15** was further tested on four species that were representative of monocotyledonous and dicotyledonous plants at a dosage of 1500 g ha$^{-1}$ located in a greenhouse. As shown in Figure 4, in most cases, the target compounds displayed stronger herbicidal activity against monocotyledonous plants than dicotyledonous plants. Moreover, it was found that most of the target compounds have stronger pre-emergent herbicidal activity than post-emergent herbicidal activity. For example, compounds **QPP-1**, **QPP-3**, **QPP-7**, and **QPP-11** exhibited good herbicidal activity against all the weeds tested with sum inhibition 167.7%, 215.7%, 259.4%, and 225.8% under pre-emergence conditions, respectively, while these compounds have lower sum inhibition (75.9%, 60.0%, 208.4%, 107.2%, respectively) under post-emergence conditions.

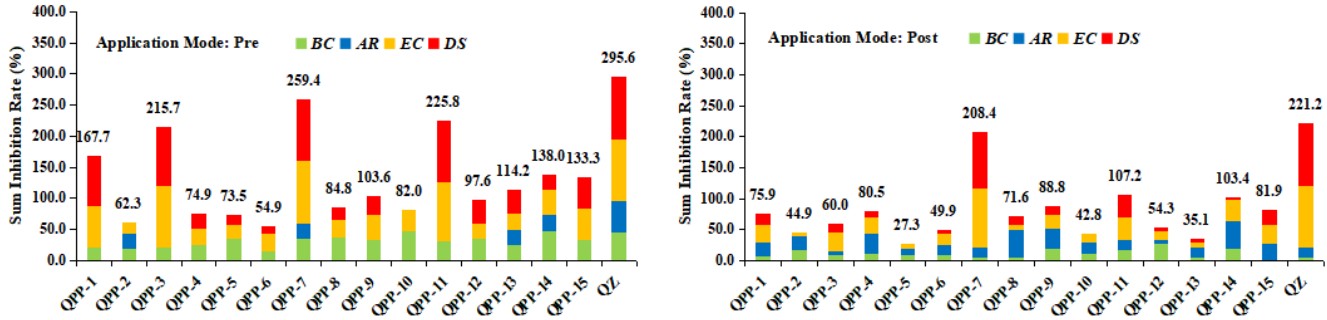

**Figure 4.** Effects (% inhibition) of compounds **QPP-1** to **QPP-15** on the loss of plant weight at a dosage of 1500 g ha$^{-1}$ in greenhouse testing; pre: pre-emergence; post: post-emergence; BC: *B. campestris*; AR: *A. retroflexus*; EC: *E. crusgalli*; DS: *D. sanguinalis*.

Analyzing the herbicidal activity of **QPP-1** to **QPP-15** under pre-emergence conditions, it was found that R group on the benzene ring of quinazolin-4-one has significant influence on the herbicidal activity. Generally, when a single substituent was introduced at the 6-position on benzene ring of quinazolin-4-one, the herbicidal activity of target compounds was improved. For example, compounds **QPP-3** (R = 6-Me, sum inhibition = 215.7%), **QPP-7** (R = 6-F, sum inhibition = 259.4%), and **QPP-11** (R = 6-Cl, sum inhibition = 225.8%) exhibited stronger herbicidal activity than that of compound **QPP-1** (R = H, sum inhibition = 167.7%). Simultaneously, the herbicidal activity was enhanced with increasing the electron-withdrawing ability of the substituent at the 6-position, i.e., R = 6-F (**QPP-7**) > 6-Cl (**QPP-11**) > 6-Me (**QPP-3**). When a single substituent (regardless of whether electron-withdrawing group or electron-donating group) was introduced at the 5-position

(i.e., **QPP-2**, **QPP-6**, **QPP-10**), or 7-position (i.e., **QPP-4**, **QPP-8**, **QPP-12**), or 8-position (i.e., **QPP-5**, **QPP-9**, **QPP-13**) on the benzene ring of quinazolin-4-one, the herbicidal activity of target compound was decreased sharply. In addition, when a substituent was introduced at the 7-position of **QPP-3** and **QPP-7**, the corresponding disubstituted compounds **QPP-14** and **QPP-15** showed lower herbicidal activity than that of **QPP-3** and **QPP-7**. These results suggested that the spatial position of R group on the benzene ring of quinazolin-4-one has more important influence on the herbicidal activity than that of electronic effect and introducing a single electron-withdrawing group at the 6-position on benzene ring of quinazolin-4-one would be essential for improving herbicidal activity.

As a result of the higher sum inhibition against all the weeds tested, compounds **QPP-3**, **QPP-7**, and **QPP-11** were chosen for further testing at lower doses. As shown in Table 1, these compounds displayed stronger herbicidal activity against monocotyledonous plants than dicotyledonous plants. Moreover, upon decreasing the dosage, the herbicidal activity of these compounds under post-emergence conditions decreased faster than that observed under pre-emergence conditions. These findings confirmed that the target compounds have better selective to monocotyledonous plants and exhibit stronger herbicidal activity under pre-emergence conditions than under post-emergence conditions. It was found that compounds **QPP-3**, **QPP-7**, and **QPP-11** exhibit good herbicidal activity against monocotyledonous plants under pre-emergence conditions at a dosage of 750 g ha$^{-1}$. Unfortunately, these compounds have lower herbicidal activity against monocotyledonous plants than that of QZ when the dosage reduced to 187.5 g ha$^{-1}$. Nevertheless, to our relief, compound **QPP-7** still exhibited excellent pre-emergent herbicidal activities against *E. crusgalli* and *D. sanguinalis* with inhibition 96.7% and 100% at the dosage of 375 g ha$^{-1}$, respectively, which are almost equal to QZ (Figure 5). This promising result indicate that compound **QPP-7** may serve as a potential lead compound for further optimization.

**Table 1.** Effects (inhibition/%) of compounds **QPP-3**, **QPP-7**, and **QPP-11** on loss of plant weight at lower dosage in greenhouse testing [a].

| Comp. | Rate (g ha$^{-1}$) | *B. campestris* | | *A. retroflexus* | | *E. crusgalli* | | *D. sanguinalis* | |
|---|---|---|---|---|---|---|---|---|---|
| | | Pre | Post | Pre | Post | Pre | Post | Pre | Post |
| **QPP-3** | 750 | 8.3 ± 1.7 | 0 | 0 | 0 | 94.5 ± 1.3 | 24.7 ± 1.6 | 88.9 ± 0.2 | 1.9 ± 0.4 |
| | 375 | 0 | 0 | 0 | 0 | 60.4 ± 1.4 | 5.0 ± 1.0 | 76.7 ± 1.3 | 0 |
| | 187.5 | 0 | 0 | 0 | 0 | 45.0 ± 2.1 | 0 | 48.9 ± 1.8 | 0 |
| **QPP-7** | 750 | 7.8 ± 0.9 | 0 | 4.8 ± 0.2 | 0 | 100 | 42.1 ± 0.6 | 100 | 61.6 ± 1.9 |
| | 375 | 0 | 0 | 0 | 0 | 96.7 ± 1.6 | 24.0 ± 0.6 | 100 | 21.3 ± 1.2 |
| | 187.5 | 0 | 0 | 0 | 0 | 64.8 ± 3.1 | 0 | 55.6 ± 1.4 | 0 |
| **QPP-11** | 750 | 4.2 ± 0.3 | 6.6 ± 0.4 | 0 | 0 | 91.2 ± 1.5 | 15.7 ± 1.3 | 84.4 ± 0.8 | 13.2 ± 0.4 |
| | 375 | 0 | 0 | 0 | 0 | 75.8 ± 0.4 | 10.2 ± 1.0 | 72.2 ± 1.1 | 6.6 ± 0.1 |
| | 187.5 | 0 | 0 | 0 | 0 | 60.4 ± 0.8 | 0 | 53.3 ± 2.0 | 0 |
| QZ | 750 | 38.6 ± 2.3 | 7.0 ± 1.1 | 31.2 ± 0.7 | 0 | 100 | 100 | 100 | 100 |
| | 375 | 24.5 ± 1.7 | 0 | 20.6 ± 1.3 | 0 | 100 | 100 | 100 | 100 |
| | 187.5 | 11.1 ± 0.6 | 0 | 10.5 ± 1.7 | 0 | 97.8 ± 1.7 | 100 | 100 | 100 |

[a] Each value represents the mean ± SD of three experiments.

In order to explore the effect of R$_1$ group on herbicidal activity, subsequently, the target compounds **QPP-16** to **QPP-29** were synthesized by using **QPP-7** as lead compound, and their herbicidal activity was evaluated under pre-emergence conditions. Lead compound **QPP-7** and QZ were selected as the positive control samples. As shown in Figure 6, the target compounds **QPP-16** to **QPP-29** showed lower sum inhibition against all the weeds tested at a dosage of 1500 g ha$^{-1}$ than that of lead compound **QPP-7** and QZ. Meanwhile, it was found that the decreased sum inhibition was mainly attributed to the reduced inhibitory effect of the compound on monocotyledonous plants. Analyzing the herbicidal activity of **QPP-7** and **QPP-16** to **QPP-18**, it was found that, with the extension of the carbon chain, compounds with longer carbon chain progressively lost herbicidal activity. When a branched-chain alkyl groups was introduced on the nitrogen atom, **QPP-19** showed lower

herbicidal activity than that of the corresponding straight-chain compound **QPP-18**. With the alkyl group at the nitrogen atom of quinazolin-4(*H*)-one replaced by a benzene ring, the herbicidal activity of **QPP-20** did not improve. Although the introduction of substituent on the benzene ring at the nitrogen atom increased the herbicidal activity compared to **QPP-20**, the herbicidal activity of **QPP-21** to **QPP-29** was still lower than that of lead compound **QPP-7**. These results suggested that the introduction of a substituent with bulker than the methyl at the nitrogen atom did not conducive to improving the herbicidal activity.

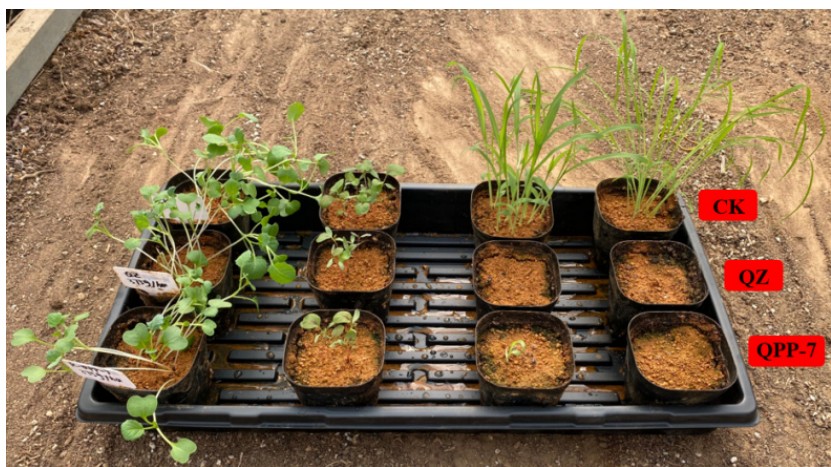

**Figure 5.** Photographs illustrating the herbicidal activity of **QPP-7** under pre-emergence conditions at a dosage of 375 g ha$^{-1}$; From left to right: *B. campestris*, *A. retroflexus*, *E. crusgalli*, *D. sanguinalis*.

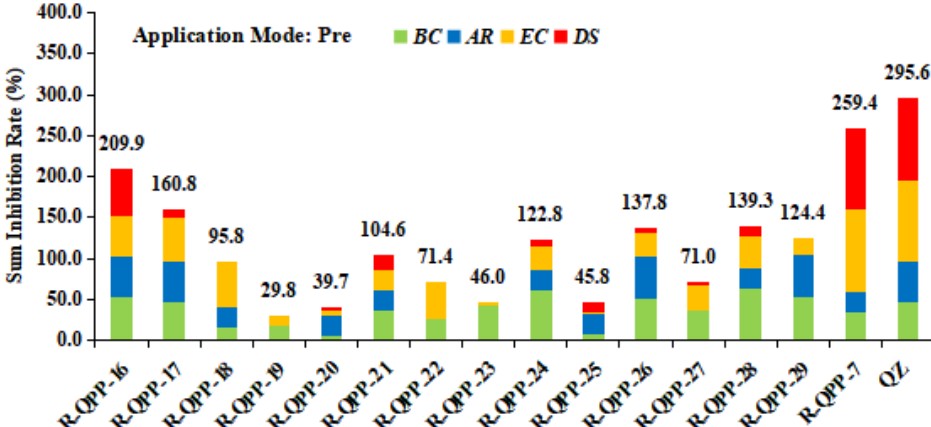

**Figure 6.** Effects (% inhibition) of compounds **QPP-16** to **QPP-29** on the loss of plant weight at a dosage of 1500 g ha$^{-1}$ in greenhouse testing; pre: pre-emergence; BC: *B. campestris*; AR: *A. retroflexus*; EC: *E. crusgalli*; DS: *D. sanguinalis*.

The aforementioned results for structure-activity relationship revealed that the herbicidal activity of target compounds is strongly influenced by the spatial position of R group and the bulk of R$_1$ group on quinazolin-4(*H*)-one, and the (R = 6-F, R$_1$ = Me) pattern was confirmed as the optimal orientation.

### 3.4. Herbicidal Spectrum and Crop Safety of Compound QPP-7

To further evaluate whether compound **QPP-7** has the potential to be developed as a herbicide, its herbicidal spectrum against monocotyledonous plants and crop safety were investigated at a dosage of 375 g ha$^{-1}$ under pre-emergence conditions. The monocotyledonous plants *Echinochloa crusgalli* (EC), *Digitaria sanguinalis* (DS), *Pennisetum alopecuroides* (PA), *Setaria viridis* (SV), *Eleusine indica* (EI), *Avena fatua* (AF), *Elymus dahuricu* (ED), *Spartina alterniflora* (SA) were chosen as the target weeds to evaluate the herbicidal spectrum of compound

**QPP-7** in a greenhouse. As shown in Figure 7, **QPP-7** displays strong control with inhibition >90% against all the weeds tested, which is almost equal to QZ. This finding indicated that compound **QPP-7** has a broad herbicidal spectrum for monocotyledonous weeds control. Subsequently, six representative crops, *Oryza sativa*, *Zea mays*, *Triticum aestivum*, *Gossypium spp*, *Glycine max* and *Arachis hypogaea*, were selected for further crop selectivity study (Table 2). The results showed that *O. sativa*, *T. aestivum*, and *A. hypogaea* displayed a high tolerance toward compound **QPP-7**, while QZ was not selective for *O. sativa* and *T. aestivum* (98.7% and 59.3% injury, respectively). These promising results indicated that compound **QPP-7** has the potential to be developed as a pre-emergence herbicide lead compound for weed control in *O. sativa*, *T. aestivum*, and *A. hypogaea* Fields.

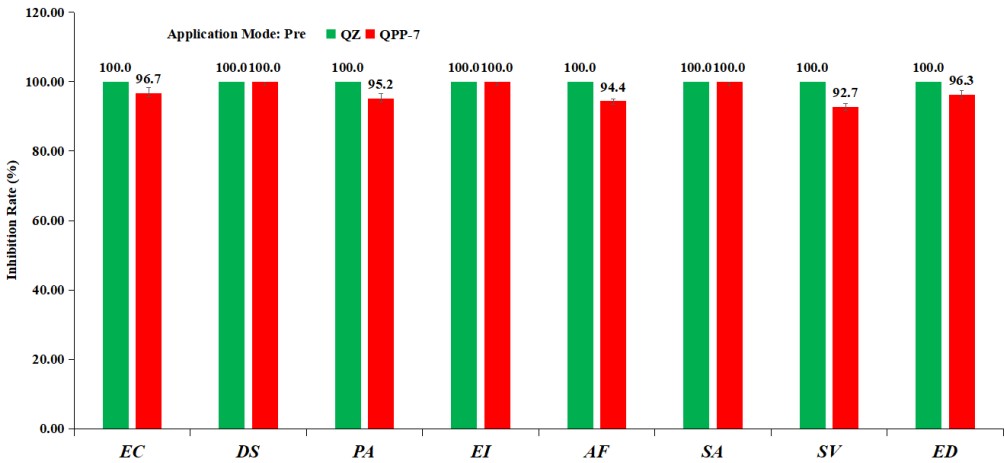

**Figure 7.** Herbicidal spectrum testing of compound **QPP-7** under pre-emergence conditions at a dosage of 375 g ha$^{-1}$; *Echinochloa crusgalli* (EC), *Digitaria sanguinalis* (DS), *Pennisetum alopecuroides* (PA), *Setaria viridis* (SV), *Eleusine indica* (EI), *Avena fatua* (AF), *Elymus dahuricu* (ED), *Spartina alterniflora* (SA).

**Table 2.** Pre-emergence crop selectivity of compound **QPP-7** at the dosage of 375 g ha$^{-1}$ (Injury Inhibition) [a].

| Comp. | % Injury | | | | | |
|---|---|---|---|---|---|---|
| | *O. sativa* | *Z. mays* | *T. aestivum* | *G. spp* | *G. max* | *A. hypogaea* |
| **QPP-7** | 0 | 55.5 ± 1.4 | 0 | 10.9 ± 1.1 | 4.5 ± 0.4 | 0 |
| QZ | 98.7 ± 0.9 | 16.8 ± 1.7 | 59.3 ± 1.2 | 0 | 0 | 0 |

[a] Each value represents the mean ± SD of three experiments.

### 3.5. Molecular Mode of Action of the Target Compound QPP-7

In order to explore the molecular mode of action of target compounds, compound **QPP-7** was selected to study the herbicidal mechanism. Since the target compounds were designed based on the APP motif, we speculated that **QPP-7** could be a ACCase inhibitor. Thus, the molecular docking simulations were first carried out. As shown in Figure 8, it was easy to find that the relevant interactions between compound **QPP-7** and the target enzyme is different to that of quizalofop-P-ethyl. In the docking complex of compound **QPP-7**, the carbonyl oxygen atom on the carboxyl group formed two non-classical hydrogen bonds with the amino acid residues of GLY1971; the carbonyl oxygen atom on the quinazolin-4(3*H*)-one formed a non-classical hydrogen bonds with the amino acid residues of GLY1997. Meanwhile, the benzene ring on the phenoxypropionic acid inserted into the active site and generated a π–π interaction with PHE1956, and the hydroxyl group generated a p–π interaction with TRP1924. In the docking complex of QZ, the hydroxyl group formed a classical hydrogen bond and two non-classical hydrogen bonds with the amino acid residues of ALA1627, GLY1626, GLY1734, respectively. Meanwhile, the benzene ring of quinoxaline formed two π–π interaction with TYR1738, and the nitrogen atom formed two non-classical hydrogen bonds with GLY1998. From the above docking results alone, it is difficult to conclude if it is **QPP-7** or QZ has a more prominent inhibitory activity

against ACCase. Therefore, ACCase activity test is necessary to carry out to help us make a accurate judgment.

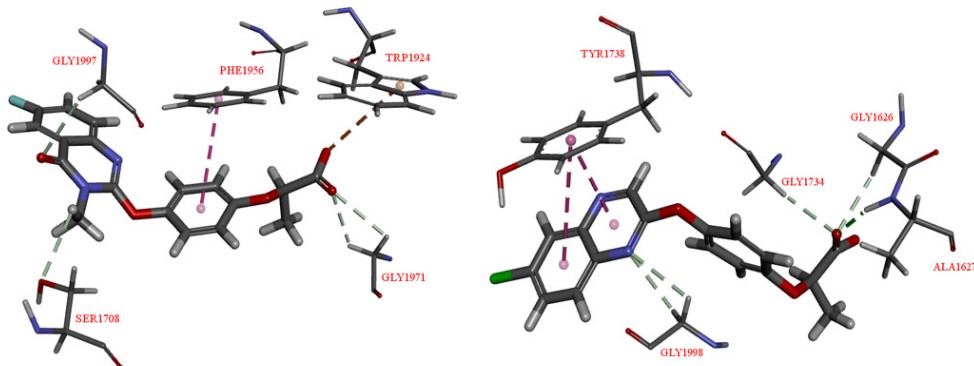

**Figure 8.** The docking binding mode of the free acid of **QPP-7** (**Left**) and quizalofop-P-ethyl (**Right**) to acetyl-CoA carboxylase (ACCase) (PDB code: 1UYR).

To further verify whether compound **QPP-7** is a ACCase inhibitor, the tests of *E. crusgalli* ACCase inhibition activity in vitro was performed. As shown in Table 3, compound **QPP-7** displayed good inhibitory activity against *E. crusgalli* ACCase with an $IC_{50}$ value of 54.65 nM, which is comparable to commercial herbicide quizalofop-P-ethyl ($IC_{50}$= 41.19 nM). This result indicate that **QPP-7** may be an ACCase inhibitor and has an herbicidal mechanism similar to that of QZ.

**Table 3.** In vitro inhibitory activity of compound **QPP-7** against *E. crusgalli* ACCase [a].

| Comp. | Regression Equation | $IC_{50}$ (nM) | 95% Confidence Interval | r |
|---|---|---|---|---|
| **QPP-7** | y = 0.4842x + 16.9 | 54.65 | 40.26 to 82.67 | 0.9751 |
| QZ | y = 0.4449x + 24.2 | 41.19 | 27.82 to 69.67 | 0.9585 |

[a] y: ACCase inhibition rate; x: concentration of tested compounds; r: Correltion coefficient; $IC_{50}$: the half maximal inhibitory concentration.

It was noteworthy that although **QPP-7** has a herbicidal mechanism similar to that of QZ, however, the study of molecular docking showed that their interactions with the ACCase is different, indicating that **QPP-7** has the potential to control weeds that are resistant to APP herbicides. Furthermore, in our present work, compound **QPP-7** has the comparable ACCase inhibitory activity to that of QZ, but its herbicidal activity at a lower dosage is worse than that of QZ. The reason may be attributed to the compound **QPP-7** with natural structure fragment being easily metabolized in plants when compared to QZ.

## 4. Conclusions

In summary, a series of quinazolin-4(3*H*)-one derivatives based on the aryloxyphenoxypropionate motif have been designed by using molecular hybridization strategy. twenty-nine novel quinazolin-4(3*H*)-one derivatives were prepared in moderate to good yields. The bioassay results showed that compound **QPP-7** displayed good pre-emergent herbicidal activity at a dosage of 375 g ha$^{-1}$. The herbicidal spectrum and crop selectivity study revealed that compound **QPP-7** had a broad spectrum of monocotyledonous weed control and displayed excellent crop safety to *O. sativa*, *T. aestivum*, and *A. hypogaea*, which indicated its great potential as a herbicide lead compound. The study of structure-activity relationship showed that the spatial position of R group and the bulk of $R_1$ group on quinazolin-4-one have strongly influenced on the herbicidal activity of target compounds, and the (R = 6-F, $R_1$ = Me) pattern was confirmed as the optimal orientation. Furthermore, the inhibitory activity against *E. crusgalli* ACCase enzyme and the molecular docking simulation of the free acid of compound **QPP-7** were performed, and the results indicated that compound **QPP-7** may be a ACCase inhibitor. For developing improved herbicidal

activity of APP herbicide containing quinazolin-4(3*H*)-one skeleton, further studies on the structural optimization of compound **QPP-7** are ongoing in our laboratory.

**Supplementary Materials:** The following supporting information can be downloaded at: https://www.mdpi.com/article/10.3390/agronomy12081840/s1, Details on the inhibition of the root growth of *B. campestris*, inhibition of the seedling growth of *E. crusgalli*, greenhouse tests, and crop selectivity, [1]H NMR, [13]C NMR and HRMS spectrum of target compounds.

**Author Contributions:** Conceptualization, K.L.; methodology, K.L.; validation, C.W., K.C. and N.L.; formal analysis, S.F. and P.L.; investigation, C.W., K.C. and N.L.; data curation, C.W. and K.C.; writing—original draft preparation, K.L.; writing—review and editing, K.L. and L.J.; project administration, K.L.; funding acquisition, K.L., X.W., G.L. and L.J. All authors have read and agreed to the published version of the manuscript.

**Funding:** This project was financially supported by the China Postdoctoral Science Foundation (No. 2020M671984); the National Natural Science Foundation of China (No. 31701827); the National Innovation and Entrepreneurship Training Program for College Students (No. 202110447022); Guangyue Young Scholar Innovation Team of Liaocheng University (No. LCUGYTD2022-04); the National Key Research and Development Program of China (No. SQ2020YFF0422322); the Natural Science Foundation of Shandong Province (No. ZR202102180037) and the Research Fund of Liaocheng University (No. 318012106).

**Institutional Review Board Statement:** Not applicable.

**Informed Consent Statement:** Not applicable.

**Data Availability Statement:** Not applicable.

**Conflicts of Interest:** The authors declare no conflict of interest.

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
