# Peer review of "Design, Synthesis, Mode of Action and Herbicidal Evaluation of Quinazolin-4(3H)-one Derivatives Based on Aryloxyphenoxypropionate Motif"

_agronomy, doi:10.3390/agronomy12081840_

Round 1
Reviewer 1 Report
Line 284 -296 Please briefly explain the procedure in the section; I understand the detailed information is given in supplementary info; however, please provide the concentration/ dose tested in the main text.
Please mention the spray specifications such as nozzle, spray pressure, etc., (please refer to any standard weed science papers)
Figure 6 please mark non-treated, solvent control, and treatments.
Author Response
Responses to the comments of reviewers:
Reviewer: 1
- Line 284 -296 Please briefly explain the procedure in the section; I understand the detailed information is given in supplementary info; however, please provide the concentration/ dose tested in the main text.
Answer:
Thank you very much for your valuable comments. We have revised the article according to your suggestion, and we have briefly explain the procedure in the main text.
- Please mention the spray specifications such as nozzle, spray pressure, etc., (please refer to any standard weed science papers)
Answer:
Thank you very much for your valuable comments.
- Figure 6 please mark non-treated, solvent control, and treatments.
Answer:
Thank you very much for your valuable comments. We have mark non-treated, solvent control, and treatments in the Figure 5 (namely,Figure 6).
Reviewer 2 Report
The topic of the study is very interesting and relevant. Weeds are known to become resistant to herbicides over time. Therefore, the search for new chemicals capable of controlling weeds will continue to be an important challenge in agronomy. The manuscript is well organized, but some things need to be corrected:
Line 345 – should be Figure 3;
Line 425 – please enlarge the letters and numbers in the figure, specify the meanings of the abbreviations;
Line 455 – please explain the interdependence of which characteristics are described by regression equations.
I made a few small notes in the manuscript.

Author Response
Responses to the comments of reviewers:
Reviewer: 2
The topic of the study is very interesting and relevant. Weeds are known to become resistant to herbicides over time. Therefore, the search for new chemicals capable of controlling weeds will continue to be an important challenge in agronomy. The manuscript is well organized, but some things need to be corrected:
- Line 345 – should be Figure 3;
Answer:
Thank you very much for your valuable comments. We have revised the article according to your suggestion.
- Line 425 – please enlarge the letters and numbers in the figure, specify the meanings of the abbreviations;
Answer:
Thank you very much for your valuable comments. We have revised the article according to your suggestion.
- Line 455 – please explain the interdependence of which characteristics are described by regression equations.
Answer:
Thank you very much for your valuable comments. Y represent ACCase inhibition rate; X represen tconcentration of tested compounds; IC50 represent the half maximal inhibitory concentration.
Reviewer 3 Report
This paper is focused on the lead discovery research of AOPP herbicide and is suitable for publication in this journal through minor revision.
QPP-7 has a fluorine atom in position 6 of the aromatic ring and the reason of its high herbicidal activity may be that fluorine atome not only has strong electronic effect, but also increases the liphophilicity of molecule.
I suggest you synthesize further new QPP derivatives possessing amide moiety instead of ester moiety. As you may know, Metamifop possessing amide moiety is another ACCase inhibitor which has strong grass weeds herbicidal activity and excellent rice safety.
Correction:
1. Page 14, line 424
pre-emergence herbicide lead compoung for weed control in O. sativa, Z. mays, T. aestivum, and A. hypogaea Field.
to
pre-emergence herbicide lead compoung for weed control in O. sativa, T. aestivum, and A. hypogaea Field.
(QPP-7 is not safe to Z. mays in Table 2.)
2. Page 16, line 461
excellent crop safety to O. sativa, T. aestivum, G, spp. and A. hypogaea
to
excellent crop safety to O. sativa, T. aestivum, and A. hypogaea
(QPP-7 is not safe to G, spp in Table 2.)
Author Response
Responses to the comments of reviewers:
Reviewer: 3
This paper is focused on the lead discovery research of AOPP herbicide and is suitable for publication in this journal through minor revision. QPP-7 has a fluorine atom in position 6 of the aromatic ring and the reason of its high herbicidal activity may be that fluorine atome not only has strong electronic effect, but also increases the liphophilicity of molecule. I suggest you synthesize further new QPP derivatives possessing amide moiety instead of ester moiety. As you may know, Metamifop possessing amide moiety is another ACCase inhibitor which has strong grass weeds herbicidal activity and excellent rice safety.
Correction:
- Page 14, line 424
pre-emergence herbicide lead compoung for weed control in O. sativa, Z. mays, T. aestivum, and A. hypogaea Field. to pre-emergence herbicide lead compoung for weed control in O. sativa, T. aestivum, and A. hypogaea Field. (QPP-7 is not safe to Z. mays in Table 2.)
Answer:
Thank you very much for your valuable comments. We have revised the article according to your suggestion.
- Page 16, line 461
excellent crop safety to O. sativa, T. aestivum, G. spp. and A. hypogaea to excellent crop safety to O. sativa, T. aestivum, and A. hypogaea (QPP-7 is not safe to G. spp in Table 2.)
Answer:
Thank you very much for your valuable comments. We have revised the article according to your suggestion.